# Revealing Drug-Target Interactions with Computational Models and Algorithms

**DOI:** 10.3390/molecules24091714

**Published:** 2019-05-02

**Authors:** Liqian Zhou, Zejun Li, Jialiang Yang, Geng Tian, Fuxing Liu, Hong Wen, Li Peng, Min Chen, Ju Xiang, Lihong Peng

**Affiliations:** 1School of Computer Science, Hunan University of Technology, Zhuzhou 412007, China; zhoulq11@163.com (L.Z.); liufxicloud@me.com (F.L.); wenh_hut@163.com (H.W.); 2School of Computer Science, Hunan Institute of Technology, Henyang 421002, China; lzjfox@163.com (Z.L.); chenmin@hnit.edu.cn (M.C.); 3Geneis (Beijing) Co. Ltd., Beijing 100102, China; yangjl@geneis.cn (J.Y.); tiang@geneis.com (G.T.); 4School of Computer Science, University of Science and Technology of Hunan, Xiangtan 411201, China; plpeng@hnu.edu.cn; 5School of Computer Science and Engineering, Central South University, Changsha 410083, China; xiangju@csu.edu.cn; 6Neuroscience Research Center, Department of Basic Medical Sciences, Changsha Medical University, Changsha 410219, China

**Keywords:** drug-target interaction prediction, computational models, network-based methods, machine learning-based methods, drug repositioning

## Abstract

Background: Identifying possible drug-target interactions (DTIs) has become an important task in drug research and development. Although high-throughput screening is becoming available, experimental methods narrow down the validation space because of extremely high cost, low success rate, and time consumption. Therefore, various computational models have been exploited to infer DTI candidates. Methods: We introduced relevant databases and packages, mainly provided a comprehensive review of computational models for DTI identification, including network-based algorithms and machine learning-based methods. Specially, machine learning-based methods mainly include bipartite local model, matrix factorization, regularized least squares, and deep learning. Results: Although computational methods have obtained significant improvement in the process of DTI prediction, these models have their limitations. We discussed potential avenues for boosting DTI prediction accuracy as well as further directions.

## 1. Introduction

Drug discovery is a complicated, costly and low-success process. It is estimated that it takes about 10∼15 years and 0.8∼1.5 billion dollars from initially presenting the abstract concept to putting it into market for a new drug. Despite pharmaceutical companies investing enormous costs and time, only about 10% of drugs are successfully evaluated by FDA every year [1,2]. Nobel Laureate James Black presented that the most solid foundation for new drug discovery is beginning from old drugs [3]. Drug repurposing, which repositions the existing drugs to find new treatment clues of the old drugs, can shorter drug research and development time, reduce unexpected drug toxicity, and promote drugs to enter clinical phases as soon as possible [4,5,6]. Jin et al. [7] represented that repositioned drugs account for about 30% of the newly FDA-approved drugs and vaccines.

DTI identification, aiming to find potential targets/drugs for the existing drugs/targets, has been an important step in drug repositioning. With the integration of numerous heterogeneous biological data, a variety of computational approaches have been exploited to systematically infer possible DTIs. Some research [4,8] has been better summarized. Inspired by these summarizations, in this study, we discussed relevant data repositories, different computational models and their advantages, and challenges for DTI identification.

## 2. Data Representation and Repositories

### 2.1. Benchmark Data Set

The majority of computational models for DTI identification used the datasets provided by Yamanishi et al. [9]. The details are shown in Table 1. Yamanishi et al. [9] provided three types of data: drug similarity matrix $S_{D}\in {ℜ}^{n\times n}$, target similarity matrix $S_{T}\in {ℜ}^{m\times m}$, and drug-target interaction network $Y\in {ℜ}^{n\times m}$ where $y_{ij}=1$ if the drug $d_{i}$ and the target $t_{j}$ is linked; otherwise, $y_{ij}=0$.

### 2.2. Flowchart

Various DTI inference algorithms have been designed over the past two decades. These methods usually integrated the datasets provided by Yamanishi et al. [9] and other biological information from various public databases into their proposed computational models, and then trained the models, finally scoring the interaction probabilities for unknown drug-target pairs. We briefly represented the flowchart as Figure 1.

### 2.3. DTI Relevant Databases

Various experimental data provides abundant information for DTI identification and significantly improved the performances of DTI prediction models. It is feasible to merge these DTI data from different databases. To address the conflict problems between data values from different repositories in the process of data merging, for example, Liu et al. [10] set a priority for each DTI and give precedence to the more reliable data source. Liu et al. [10] merged different compound-protein interaction data retrieved from Matador, DrugBank, and STITCH. Matador and DrugBank are manually curated databases. STITCH is a comprehensive repository collected from four different sources: manually curated databases, experimental validation, text mining and model prediction. Particularly, STITCH assigns each DTI a score ranging from 0 to 1000. Each score indicates confidence degree of each DTI supported by the above four types of evidence. In addition, Liu et al. [10] considered that DTIs from Matador and DrugBank are supported by biochemical experiments and the literature and gave these DTIs the highest score of 1000.

Lou et al. [11] designed a novel Network integration pipeline for DTI prediction, DTINet. DTINet developed other ways of DTI data merging from a multiple-views perspective based on the following steps:Step 1Extracting related data from different databases: (i) drugs, DTIs and drug-drug interactions from DrugBank; (ii) proteins, and protein-protein interactions from HPRD [12]; (iii) diseases, drug-disease and protein-disease associations from the Comparative Toxicogenomics Database [13]; (iv) side-effects and drug-side-effect associations from SIDER [14].Step 2Excluding isolated entities (nodes) which have no edges in the network.Step 3Integrating four types of nodes and six types of associations (edges) in Step 1 and constructing a heterogeneous network.Step 4Building multiple similarity networks to further increase the network heterogeneity.Step 5Removing homologous proteins or similar drugs from constructed heterogeneous networks to reduce the potential redundancy in the DTIs: (i) removing the DTIs involving homologous proteins with sequence identity scores larger than 40%; (2) removing the DTIs involving similar drugs with Tanimoto coefficients larger than 60%; (3) removing the DTIs involving the drugs with Jaccard similarity scores of side effects larger than 60%; (4) removing the DTIs involving the proteins or drugs associated with similar diseases (Jaccard similarity scores larger than 60%; (5) removing the DTIs involving either homologous proteins with sequence identity scores larger than 40% or similar drugs with Tanimoto coefficients larger than 60%.

Parts of DTI repositories are described as follows:

#### 2.3.1. DrugBank

The DrugBank database [15] (https://www.drugbank.ca/) provides 12,701 drug entries, which includes 2536 FDA-approved small molecules, 1279 FDA-approved biotech drugs, 130 nutraceuticals and more than 5822 experimental drugs. DrugBank describes drug details including chemical structures, pharmacological and pharmaceutical information. Further, DrugBank provides 5144 non-redundant proteins linking these drug entries and protein details including sequences, structures and pathways.

#### 2.3.2. SuperTarget

The SuperTarget database [16] (http://insilico.charite.de/supertarget/) provides comprehensive data services and links with nine websites: BindingDB, RCSB PDB, PubChem, UniProt, KEGG, DrugBank, SuperCyp, SIDER, and ConsensuspathDB. It consists of six different types of entities: drugs, proteins, side-effects, pathways, ontologies and a special subgroup of targets (the cytochromes 450). The database contains 332,828 interactions between 6219 proteins and 195,770 compounds. The drugs can be searched by drug name, PubChem ID, ATC-code, and side-effects. The targets can be retrieved by target name, EC-number, UniProt name, accession number, PDB ID, and KEGG target ID. In addition, SuperTarget provides 282 drug-target-related pathways, 6532 drug-target-related ontologies, and 63 cytochromes.

#### 2.3.3. STITCH

The STITCH database [17] (http://stitch.embl.de/) contains 1.6 bn interactions between 0.5 million chemicals and 9.6 million proteins from 2031 organisms.

#### 2.3.4. ZINC

The ZINC database [18] (http://zinc.docking.org/) is a free and curated collection of commercially-available compounds for virtual screening. It provides more than 350,000,000 purchasable compounds in ready-to-dock and 3D formats and can search compounds by one or more ZINC IDs, a specific target or targets, component ring names, common compound names, CAS/MDL Number, and vendor or catalog-specific code.

#### 2.3.5. IUPHAR/BPS Guide to PHARMACOLOGY

IUPHAR/BPS Guide to PHARMACOLOGY [19] (http://www.guidetopharmacology.org/) is an open-access website and provides an expert-driven guide to pharmacological targets. The website provides interaction information between 9459 ligands and 2917 targets. Ligands contains FDA-approved drugs, synthetic organics, inorganics, antibodies, labelled ligands, metabolites, natural products, endogenous peptides and other peptides. Targets contains G protein-coupled receptors, ion channels, nuclear hormone receptors, kinases, catalytic receptors, transporters, enzymes, and other protein targets.

#### 2.3.6. SIDER

The SIDER database [20] (http://sideeffects.embl.de/) contains 139,756 drug-side effect pairs between 5868 side effects and 1430 drugs.

#### 2.3.7. BindingDB

BindingDB [21] (http://www.bindingdb.org/bind/index.jsp) is a web-accessible database mainly focusing the interactions between drugs and proteins which can be candidate drug targets acting on small, drug-like molecules. It contains 1,558,402 binding data between 697,594 small molecules and 7233 protein targets. In addition, it provides 2291 protein-ligand crystal structures for proteins with 100% sequence identity and 5816 crystal structures for proteins with 85% sequence identity.

#### 2.3.8. TTD

Therapeutic Target Database (TTD) [22] (https://db.idrblab.org/ttd) is an open-access website that can download different types of biological information including drug structure, therapeutic targets, pathway information, and drug combinations. The database provides 2104 drug resistance mutations targeting 63 diseases, 758 targets from 12,615 patients of 70 diseases, 629 targets across various tissues from 2565 healthy individuals, 2612 target combination, and 25,333 multi-target drugs.

#### 2.3.9. MATADOR

The MATADOR database [16] (http://matador.embl.de/) is a manually annotated chemical-protein interaction website. The database differs from other resources in that it provides any direct and indirect interactions between chemical and proteins assembled by automated text-mining and manual curation. Each interaction can be deduced by retrieving the PubMed or OMIM database.

#### 2.3.10. ChEMBL

The ChEMBL database [23] (https://www.ebi.ac.uk/chembl/) is a open-access website about bioactive drug-like small molecules. It provides 15,207,914 activities from 2,275,906 compound records and 12,091 targets. The properties of small molecular drugs contain 2-D structures, calculated properties including logP, molecular weight, and Lipinski parameters, and abstracted bioactivities including binding constants, pharmacology, and ADMET.

#### 2.3.11. DCDB

The DCDB database [24] (http://www.cls.zju.edu.cn/dcdb/) summarizes action pattern of coordinated drugs and provides a theoretical basis for modeling and simulating beneficial drug combinations. It contains 1363 drug combinations between 904 individual drugs and 805 targets. Furthermore, it provides three types of relevant information: combined activity/indications, drug-drug interactions, and possible mechanism for each drug combination; chemical, pharmaceutical and pharmacological properties, and known molecular targets for each drug; sequence, function and affiliated pathway for each drug target.

### 2.4. DTI Relevant Software Packages

Drug and target features are important for unknown DTI classification. Researchers have developed various software packages to extract abundant drug and target features.

#### 2.4.1. RDKit

RDKit [25] (http://www.rdkit.org/) is an open-source cheminformatics software and descriptor generator for machine learning. The website can compute various features including canonical SMILES, 2D depiction, fingerprinting, chemical reactions, molecular serialization, similarity/diversity picking, and 2D and 3D descriptions for drug molecules. The software is continuously updating from 2012.

#### 2.4.2. ChemDes

ChemDes [26] (http://www.scbdd.com/chemdes/) is a freely available web-based platform. The platform integrates multiple state-of-the-art packages including CDK, RDKit, PaDEL, Pybel, Chemopy, BlueDesc, and jCompoundMapper to compute molecular descriptors and fingerprints. It provides three convenient auxiliary tools for fingerprint similarity calculation, MOPAC optimization, and format converting. Currently, it can compute 3679 molecular descriptors and 59 types of fingerprints.

#### 2.4.3. OpenBabel

OpenBabel [27] (http://openbabel.org/wiki/Main_Page) is a open chemical toolbox. The software allows anyone to search, store, analyze, or convert data from various areas including molecular modeling and biochemistry. In addition, it can read, write, and convert over 110 chemical file formats. The software is continuously updating since 2007.

#### 2.4.4. Rchemcpp

Rchemcpp [28] (http://shiny.bioinf.jku.at/Analoging/) is an efficient web server to find structural analogs in Drugbank, ChEMBL, and Connectivity Map. These structural analogs are molecular compounds similar to a query compound. Molecule kernels are applied to compute structural similarity based on shared substructures between molecules. Rchemcpp provides various important applications for drug development, for example, prioritizing molecular compounds after screening and reducing adverse side effects in the process of late research and development.

#### 2.4.5. PyDPI

PyDPI [29] (https://sourceforge.net/projects/pydpicao/) is a comprehensive platform for separately compute features of proteins and drugs from amino acid sequences and chemical structures. It provides 42 descriptor types composed of 9890 descriptors for proteins, 13 descriptor types composed of 615 descriptors for drugs. In addition, the platform provides seven molecular fingerprint systems for drugs, including atom pair fingerprints, topological fingerprints, topological torsion fingerprints, electro-topological state fingerprints, Morgan/circular fingerprints, MACCS keys, and FP4 keys.

#### 2.4.6. Rcpi

Rcpi [30] (http://bioconductor.org/packages/release/bioc/html/Rcpi.html) is a freely available molecular informatics toolkit for finding compound-protein interactions. The toolkit is applied to represent complex molecules from proteins and drugs and complex interactions including compound-protein and protein-protein interactions. It can also compute abundant physicochemical and structural features of proteins from amino acid sequences, molecular descriptors of small molecular compounds from their structures, compound-protein interaction and protein-protein interaction descriptors.

#### 2.4.7. KeBABS

KeBABS [31] (http://www.bioinf.jku.at/software/kebabs/) is an R package to analyze biological sequences including amino acid, DNA, and RNA sequences. It complements some important kernels for sequence analysis based on kernel methods. It can efficiently select hyperparameters by cross validation (CV), nested CV and features grouped CV.

#### 2.4.8. PROFEAT

PROFEAT [32] (http://bidd2.nus.edu.sg/cgi-bin/profeat2016/main.cgi) is a open web server and can extract protein features from network properties of protein-protein interaction network and amino acid sequences. It groups physicochemical and commonly-used structural features into six categories composed of 10 features: protein, protein structure, protein-protein interaction pair, protein-ligand interaction pair, small molecule, and biological network. The calculated features include 51 descriptors and 1447 descriptor values, such as amino acid composition, dipeptide composition, Geary autocorrelation, Moran autocorrelation, normalized Moreau-Broto autocorrelation, quasi-sequence-order descriptors and composition, sequence-order-coupling number, transition and distribution of different structural and physicochemical properties. Particularly, the server can also calculate other autocorrelation descriptors with the properties defined by users. The server is always updating.

#### 2.4.9. Pse-in-One

Pse-in-One [33] (http://bioinformatics.hitsz.edu.cn/Pse-in-One/download/) is a flexible web server to effectively capture key features of a biological sample (such as protein, DNA, and RNA) from its sequence. It can generate nearly all the possible features for protein, DNA, and RNA through 28 different modes. In addition, it can also generate feature vectors based on user-defined properties.

#### 2.4.10. ProtrWeb

ProtrWeb [34] (http://protrweb.scbdd.com/) is a R package providing various numerical representation of proteins and peptides from amino acid sequences. The package provides eight descriptor categories composed of 22 types of descriptors which include about 22,700 descriptor values. It can also automatically construct customized descriptors with used-defined properties.

### 2.5. On-Line Tools/Web-Service for DTI Prediction

Stimulated by the increasing interest in DTI identification and the availability of various open data repositories, many online tools have been exploited to find new DTIs. These tools have been provided without considering the mathematical models and computational complexity, and thus significantly lower the collaboration barriers among different researchers involved in multiple disciplines. More online tools are described as follows [35].

#### 2.5.1. DrugE-Rank

DrugE-Rank [36] (http://datamining-iip.fudan.edu.cn/service/DrugE-Rank) nicely combines the advantages of feature-based and similarity-based methods with ensemble learning. Its performance is thoroughly validated by three types of main experiments on FDA approved drugs from DrugBank: cross-validation on the drugs before March 2014, independent test on the drugs after March 2014, and independent test on FDA experimental drugs.

#### 2.5.2. DINIES

DINIES [37] (http://www.genome.jp/tools/dinies/) provides integrative analyses by combing various types of heterogeneous data, for example, chemical structures and side effects of drugs, amino acid sequences and domains of target proteins. It can accept any precalculated similarity values of drugs and targets. Users can select different parameters in the supervised learning model and specify weights to integrate different heterogeneous biological data.

#### 2.5.3. Drug2Gene

Drug2Gene [38] (http://www.drug2gene.com) integrates DTI data from 19 public databases. It provides 4,372,290 unified DTIs for targets, most of which contain reported bioactivity data. It aims mainly at finding tool compounds interacting with a given target protein or identifying all known target proteins for a drug.

#### 2.5.4. iGPCR-Drug

iGPCR-Drug [39] (http://www.jci-bioinfo.cn/iGPCR-Drug/) is a sequence-based classifier to infer the associations between drugs and GPCRs in cellular networking. The high throughput tool formulates a drug compound by a 256 vector, a GPCR by pseudo amino acid composition and then predict possible drug-GPCR associations based on fuzzy *k*-nearest neighbor method.

#### 2.5.5. SynSysNet

SynSysNet [40] (http://bioinformatics.charite.de/synsysnet) is an online platform to create a comprehensive four-dimensional network from 1000 synapse specific proteins and their small molecules. It provides numerous DTI information for 750 FDA approved drugs and 50,000 compounds. Approximately 200 pathways involved can be applied to explore DTIs.

#### 2.5.6. SDTNBI

SDTNBI [41] (http://lmmd.ecust.edu.cn/methods/sdtnbi/) prioritizes possible target proteins for new chemical entities, failed drugs, and old drugs. It uses four benchmark datasets including kinases, GPCRs, nuclear receptors, and ion channels.

#### 2.5.7. DTome

DTome [42] (http://bioinfo.mc.vanderbilt.edu/DTome) extracts and integrates four types of interaction data including drug interactions, drug-gene associations, DTIs, and target-/gene-protein interactions. It utilizes web-based query method to find drug candidates and build a DTome network based on four types of interaction data. Additionally, it can analyze and interpret a DTome network based on network analysis and visualization procedures.

#### 2.5.8. PharmMapper

PharmMapper [43] (http://lilab.ecust.edu.cn/pharmmapper/) provides various repertoire of pharmacophore database related to targets in DrugBank, BindingDB, TargetBank, and possible drug target databases. The pharmacophore database contains more than 7000 receptor-based pharmacophore models. PharmMapper can automatically find the best position for a query molecule based on the models and list the top *N* best-fitted hits with similar target annotations.

#### 2.5.9. SwissTargetPrediction

SwissTargetPrediction [44] (http://www.swisstargetprediction.ch) can accurately find the target proteins of bioactive small molecules by combining 2D and 3D similarities with known ligands. It can predict protein-small molecule interactions in five different organisms including human, mouse, rat, horse and cow.

#### 2.5.10. TargetNet

TargetNet [45] (http://targetnet.scbdd.com) can find the activity for a query molecule across 623 human target proteins based on multi-target structure activity relationship analysis. It generates a DTI profiling as a feature vector of drugs to infer drug model of action, drug-drug interactions, toxicity classification, and target candidates.

#### 2.5.11. DT-Web

DT-Web [46] (http://alpha.dmi.unict.it/dtweb/) computes recommendations for a query drug combined with domain-specific knowledge representing drug and target similarities. It can find drugs acting simultaneously on multiple target proteins in a multi-pathway environment. The platform is periodically synchronized with the DrugBank database and updated accordingly.

## 3. Network-Based Methods

Computational methods for DTI prediction can be roughly classified into four categories: ligand-based approaches, docking approaches, network-based approaches, and machine learning-based approaches. Ligand-based approaches assume that similar drugs tend to bind similar targets and predict underlying DTIs based on ligand similarities. However, prediction accuracies of ligand-based approaches may be unreliable when known ligands for a protein are not enough. Docking approaches fully utilize the 3D structures of proteins, however, this type of method cannot find new DTIs when the 3D structures of proteins are unknown. Network-based approaches and machine learning-based approaches tend to address the limitations of the above two types of methods. Network-based methods efficiently predicted potential DTIs by integrating graph-based techniques and various biological data.

### 3.1. DSSI

Campillos et al. [47] exploited a drug side-effect similarity-based inference method (DSSI). DSSI can be classified into three steps:

Step 1: Developing a measure to compute the probability that two drugs share a common target based on drugs’ chemical similarity (2D Tanimoto coefficient, *y*):(1)$$P_{2D}\left(y\right)={\left(1+e^{\frac{B-y}{A}}\right)}^{-1}$$

Step 2: Measuring the probability that two drugs simultaneously interact with a target based on their phenotypic side-effect similarity(*x*):(2)$$P_{SE}\left(x\right)=A\cdot x+B$$

Step 3: Designing a sigmoid function to compute the probabilities of two drugs sharing the same target incorporating chemical similarities and phenotypic side-effect similarities.
(3)$$\begin{array}{c} P_{SE,2D}\left(x,y\right)=H\cdot {\left(1+e^{A+Z-\sqrt{{\left(C\cdot y\right)}^{2}+{\left(1-x\right)}^{2}}}\right)}^{-1}\\ Z=D\cdot {\left(\arctan\frac{y}{1-x}\right)}^{E}\cdot (B+F\cdot {\left(\arctan\frac{y}{x}\right)}^{G} \end{array}$$
where the fitted parameters A=0.0167,B=55.507,C=−810.16,D=−129.6,E=455.6,F=617.3,G=0.415,H=0.8

DSSI can find possible DTIs, however, it can only be used to infer potential associations for drugs that have known side-effect information, thus seriously limiting its application.

### 3.2. MTOI

Yang et al. [48] exploited a robust computational model to mine new drug targets based on multiple target optimal intervention solutions (MTOI). MTOI is classified into two stages: drug target identification and optimal multi-target control solution inference. In stage 1, MTOI firstly defined the disease state combing experimental data from patients and cells in abnormal conditions, and the desired state that could be restored into normal physiological state; it then selected activities of potential drug targets and calculated median deviation (m.d.) of the activities between the normal and disease states to score underlying drug targets. In stage 2, MTOI added drug reactants to screened drug targets and obtained multi-target intervention solution by selecting intensities. MTOI identified underlying drug targets and best restored an inflammation-related network to a normal state. Figure 2 described the details.

### 3.3. NRWRH

Chen et al. [49] assumed that similar drugs intend to interact with similar targets and presented a method, Network-based Random Walk with Restart on the Heterogeneous network (NRWRH) by integrating drug similarity network, protein similarity network, and DTI network into a heterogeneous network. NRWRH computed the interaction probabilities for unknown drug-target pairs by randomly walking on the heterogeneous network:(4)$$M=\left[\begin{array}{cc} M_{TT} & M_{TD}\\ M_{DT} & M_{DD} \end{array}\right]$$
NRWRH finally defined the following iteration model to compute the interaction probability by randomly walking in DTI network:(5)$$\begin{array}{c} p_{t+1}=\left(1-\gamma \right)M^{T}p_{t}+\gamma p_{0} \end{array}$$
Figure 3 describes the details.

where λ and γ is the probability of jumping from target/drug network to drug/target network and the restart of walking at the seed nodes, respectively.

### 3.4. DBSI, TBSI, and NBI

Cheng et al. [50] viewed a DTI network as a bipartite graph and developed three DTI prediction methods: Drug-based similarity inference (DBSI), Target-based similarity inference (TBSI), and Network-based inference (NBI). DBSI assumed that a query drug $d_{i}$ similar to known drugs interacting with a target $t_{j}$ may associate with $t_{j}$ and defined a linkage score between $d_{i}$ and $t_{j}$:(6)$$V_{ij}^{D}=\frac{\sum_{l=1,l\neq i}^{n}S_{D}\left(d_{i},d_{l}\right)y_{ij}}{\sum_{l=1,l\neq i}^{n}S_{D}\left(d_{i},d_{l}\right)}$$

TBSI assumed that a query target $t_{i}$ similar to known targets, which interacts with a drug $d_{i}$, may associate with $d_{i}$ and defined a linkage score between $d_{i}$ and $t_{j}$:(7)$$V_{ij}^{T}=\frac{\sum_{l=1,l\neq j}^{m}S_{T}\left(t_{j},t_{l}\right)y_{ij}}{\sum_{l=1,l\neq j}^{m}S_{T}\left(t_{j},t_{l}\right)}$$

Given a target $t_{j}$, NBI defined its score associated with $d_{i}$:(8)$$f\left(i\right)=\sum_{l=1}^{m}\frac{y_{il}}{k(t_{l})}\sum_{o=1}^{n}\frac{y_{ol}f_{0}\left(o\right)}{k(d_{o})}$$
where $f_{0}\left(o\right)=y_{oj},\;\;o\in 1,2,\ldots ,n$ is initial score of drug $d_{o}$, $k\left(d_{o}\right)=\sum_{s=1}^{m}y_{os}$ is the number of targets interacting with $d_{o}$, and $k\left(t_{l}\right)=\sum_{s=1}^{n}y_{sl}$ is the number of drugs associating with $t_{l}$.

### 3.5. DTINet

Luo et al. [11] integrated various information from multiple heterogeneous networks and presented a novel Network integration pipeline for DTI prediction, DTINet. DTINet used a compact feature learning method to handle the noisy, high-dimensional and incomplete natures of large-scale biological data and obtained low-dimensional but informative vector representations of drugs and targets. Figure 4 described the details.

## 4. Machine Learning-Based Methods

The researchers exploited numerous models and algorithms to find missing DTIs based on machine learning methods except for network-based methods. These methods can be roughly classified into five groups: Bipartite Local Model (BLM), regularized least squares, matrix factorizations, deep learning, and other methods.

### 4.1. BLM

#### 4.1.1. KRM

Yamanishi et al. [9] exploited a Kernel Regression Method (KRM). KRM scored the interaction likelihoods for unknown drug-target pairs through three stages: constructing pharmacological space, learning model based on kernel regression to represent the correlation between chemical/genome space and pharmacological feature space, and calculating feature-based similarity scores. Figure 5 describes the details.

where weight $w_{i}$ can be computed by optimizing the following loss function:(9)$$L=||UU^{T}-SWW^{T}S^{T}{\left|\right|}_{F}^{2}$$

#### 4.1.2. BLM

Bleakley et al. [51] proposed a supervised learning-based Bipartite Local Model (BLM) to find novel linkage between drug $d_{i}$ and target $t_{j}$ in the following way:

Step 1: Excluding target $t_{j}$. For a drug $d_{i}$, listing all other known targets in the bipartite network and giving their labels +1; listing the targets unknown to be targeted by $d_{i}$ and giving their labels −1.

Step 2: Finding a classification rule to discriminate the +1-labeled data from the −1-labeled data based on genomic sequence information for the targets.

Step 3: Taking this rule and identifying the label of $t_{j}$ and thus inferring whether there exists linkage between $d_{i}$ and $t_{j}$.

Step 4: Fixing the same target $t_{j}$ and excluding drug $d_{i}$, listing all other known drugs interacting $t_{j}$ in the bipartite network and giving their labels +1; listing the drugs unknown to interact with $t_{j}$ and giving their labels −1.

Step 5: Finding a classification rule to discriminate the +1-labeled data from the −1-labeled data based on chemical structure information for the drugs.

Step 6: Taking this rule and identifying the label of $t_{j}$ and thus inferring whether there exists linkage between $d_{i}$ and $t_{j}$.

Bleakley et al. [51] used SVM as local classifier.

#### 4.1.3. BLM-NII

Mei et al. [52] incorporated Neighbor-based Interaction-profile Inferring model (NII) into the BLM to find potential DTIs, especially for new drugs and targets (BLM-NII). BLM-NII can be grouped into five steps: computing NII, computing drugs and targets similarity matrix, learning a local model, computing the interaction probability, and obtaining final results. Figure 6 describes the details.

### 4.2. Regularized Least Squares

#### 4.2.1. LapRLS, NetLapRLS

Xia et al. [53] designed Laplacian regularized least squares (LapRLS) and LapRLS incorporating DTI network (NetLapRLS) to identify underlying DTIs based on a data-dependent manifold regularization model. The details are described in Figure 7.

where $K_{d}\in R^{n\times n}$ and $K_{p}\in R^{m\times m}$ represented two undirected graphs of drug domains and protein domains including both labeled and unlabeled samples, respectively.

#### 4.2.2. RLS${}_{GIP}$

Van et al. [54] assumed that a drug, which exhibits a similar interaction pattern or non-interaction pattern with targets in a known DTI network, is likely to exhibit similar interacting behavior when finding new targets for the drug. Similarly, targets have similar features. Based on the assumption, Van et al. [54] exploited a Regularized Least Squares (RLS) method combined with Gaussian Interaction Profile kernel (RLS${}_{GIP}$). RLS${}_{GIP}$ predicted new DTIs based on three steps: separately computing GIP kernels of drugs and targets, obtaining $K_{chemical,d}$ and $K_{genomica,t}$ by adding a small multiple of an identity matrix and integrating the two kernels into GIP kernel, and predicting DTIs based on RLS classifier. Figure 8 describes the details.

#### 4.2.3. WNN, WNN-GIP

Van et al. [55] developed a weighted nearest neighbor (WNN) method to infer association candidates for new drugs/targets. WNN defined an interaction profile score $y_{WNN}^{d}$ for a new drug *d* as follows:(10)$$y_{WNN}^{d}=\sum_{i=1}^{n}w_{i}y_{i}$$
where the weight $w_{i}$ can be computed by a given decay value T≤1 as $w_{i}=T^{i-1}$.

WNN [55] then extended GIP [54] with WNN and exploited WNN-GIP to identify possible association information for new drugs (or targets): for a new drug *d*, WNN-GIP add $y_{WNN}^{d}$ as a new row to original DTI matrix *Y* and apply GIP to obtain interaction profile of *d*.

#### 4.2.4. Kron-RLS

Pahikkala et al. [56] presented a Kronecker Regularized Least-Square-based method (Kron-RLS) to score unknown drug-target pairs. Given a training set *X* ($x_{i}\in X$ is a drug-target pair) and their real labels $y_{i}$ ($y_{i}=1$, if the drug interacts with the target in $x_{i}$; $y_{i}=0$, otherwise), Kron-RLS formulated the problem of DTI prediction as minimizing the following objective function:(11)$$J\left(f\right)=\sum_{i=1}^{m}(y_{i}-f\left(x_{i}\right){)}_{2}+{\lambda ||f||}_{k}^{2}$$
where ${\left||f|\right|}_{k}^{2}$ is the norm of *f*. By representation theorem, the minimization of the above function can be described as:(12)$$f\left(x\right)=\sum_{i=1}^{m}a_{i}k(x,x_{i})$$
where $a_{i}$ can be computed by the following equation:(13)(K+λI)a=y
where $K=K_{d}⊗K_{t}$ included all drug-target pairs, $K_{d}$ and $K_{t}$ represented kernel matrix of drugs and targets in the training set.

#### 4.2.5. KMDR

Kuang et al. [57] assumed that two similarity entities tend to link similar nodes to each other and developed a kernel matrix dimension reduction method (KMDR). KMDR defined a general formulation:(14)$$vec\left(\overset{\wedge }{Y}\right)=V\tilde{\Lambda }V^{T}vec\left(Y\right)$$
where vec(Y) is a drug-target pair vector, $\overset{\wedge }{Y}$ is predicted drug-target association score matrix. $K=V\tilde{\Lambda }V^{T}$ is a kernel matrix. KMDR exploited three independent sub-algorithms: KMDR-KP, KMDR-KS, and KMDR-avg.

KMDR-KP defined *K* as $K=S_{d}⊗S_{t}$ where $S_{d}=V_{d}\Lambda_{d}V_{d}$, $S_{t}=V_{t}\Lambda_{t}V_{t}$, $V=V_{d}⊗V_{t}$, $\Lambda =\Lambda_{d}⊗\Lambda_{t}$, and scored the interaction probabilities for unknown drug-target pairs by the following equation:(15)$$\overset{\wedge }{Y}=V_{d}Z^{T}V_{t}^{T}$$
where $vec\left(Z\right)=\tilde{\Lambda }vec\left(V_{t}^{T}Y^{T}V_{d}\right)$, and $\tilde{\Lambda }$ is a diagonal matrix of Λ.

KMDR-KS are similar to KMDR-KP but $\Lambda =\Lambda_{d}⊕\Lambda_{t}$.

KMDR-avg defined two kernels: $K_{d}=S_{d}$ and $K_{t}=S_{t}$, scored for unknown drug-target pairs based these two kernels, respectively:(16)$$\overset{\wedge }{Y}{}_{d}=V_{d}\tilde{\Lambda }{}_{d}V_{d}^{T}Y$$

(17)$$\overset{\wedge }{Y}{}_{t}=V_{t}\tilde{\Lambda }{}_{t}V_{t}^{T}Y^{T}$$

The final scores can be calculated as:(18)$$\overset{\wedge }{Y}=\left(V_{d}\tilde{\Lambda }{}_{d}V_{d}^{T}Y+Y^{T}{V_{t}}^{T}\tilde{\Lambda }{}_{t}V_{t}\right)/2$$

### 4.3. Matrix Factorization

As shown in Figure 9, matrix factorization methods can be used to complete the missing values in DTI matrix. The type of method first factorized *Y* into two matrices $A\in {ℜ}^{n\times k}$ and $B\in {ℜ}^{m\times k}$ satisfying $AB^{T}\approx Y$, where *A* and *B* represented latent feature vectors of drugs and targets. *k* is the number of features, k≪n,m, respectively.

#### 4.3.1. KBMF2K

Gönen [58] took DTI prediction as a binary classification problem and developed a Kernelized Bayesian Matrix Factorization with twin Kernels (KBMF2K). KBMF2K integrated three different experimental settings into a single unified framework: (i) finding interacting targets from *B* for a new drug $d_{new}$, (ii) finding interacting drugs from *A* for a new target $t_{new}$, (iii) estimating potential associations between a new drug $d_{new}$ and a new target $t_{new}$.

KBMF2K designed a deterministic variational approximation method based on fully conjugate probabilistic model and projected drugs and targets into a unified subspace. Figure 10 illustrates the proposed probabilistic model.

where ∧ and $P_{d}$ represented priors and projection matrices for a chosen subspace dimensionality, respectively. The drug kernel matrix $K_{d}$ is applied to project the drug-target pairs to a low-dimensional space, $G_{d}$ consisted of the low-dimensional feature representations of drugs. Similarly, $G_{t}$ can be computed. Finally, the predicted interaction matrix $Y_{p}$ can be calculated based on $G_{d}$ and $G_{t}$.

#### 4.3.2. PMF

Cobanoglu et al. [59] developed a probabilistic matrix factorization method (PMF) based on collaborative filtering algorithm. Using a probabilistic model with Gaussian noise, PMF defined the conditional probability for each observed interaction as follows:(19)$$p\left(Y|A,B,\sigma^{2}\right)=\prod_{i=1}^{n}\prod_{j=1}^{m}[f(y_{ij}\left|a_{i}b_{j}^{T},\sigma^{2}\right){]}^{I_{ij}}$$
where $f(y_{ij}\left|a_{i}b_{j}^{T},\sigma^{2}\right)$ denotes the Gaussianly distributed probability density function for $y_{ij}$, with mean μ and variance σ, $I_{ij}$ is an indicator function equal to 1 if $y_{ij}$ is known and 0 otherwise.

Using zero-mean, PMF represents spherical Gaussian priors on *A* and *B* as:(20)$$p\left(A|\sigma_{A}^{2}\right)=\prod_{i=1}^{N}f(a_{i}|0,\sigma_{A}^{2}I)$$
(21)$$p\left(B|\sigma_{B}^{2}\right)=\prod_{i=1}^{N}f(b_{i}|0,\sigma_{B}^{2}I)$$
PMF then computed the log-likelihood of *A* and *B*:(22)$$\begin{array}{c} \ln\left(p\left(A,B|Y,\sigma^{2},\sigma_{A}^{2},\sigma_{B}^{2}\right)\right)=-\frac{1}{2\sigma^{2}}\sum_{i=1}^{n}\sum_{j=1}^{m}I_{ij}{\left(Y_{ij}-a_{i}b_{j}^{T}\right)}^{2}-\frac{1}{2\sigma_{A}^{2}}\sum_{i=1}^{n}a_{i}a_{i}^{T}-\frac{1}{2\sigma_{B}^{2}}\sum_{j=1}^{n}b_{j}b_{j}^{T}+C \end{array}$$

Finally, the underlying DTI score matrix can be computed:(23)$$Y_{p}=AB^{T}$$

#### 4.3.3. MSCMF

Zheng et al. [60] proposed a Multiple Similarities Collaborative Matrix Factorization (MSCMF) method by integrating matrix factorization, collaborative filtering and relevant biological information including chemical structures and ATC codes of drugs and genomic sequence, GO and protein-protein interaction network of targets. MSCMF found possible DTIs based on the following seven steps:

Step 1: Building an objective function to minimize the squared error between *Y* and *A* and *B*:(24)$$\arg\min_{A,B}||Y-AB^{T}{\left|\right|}_{F}^{2}$$

Step 2: Introducing a weighted low-rank approximation model to distinguish labeled drug-target pairs from unlabeled pairs:(25)$$\arg\min_{A,B}||W\cdot \left(Y-AB^{T}\right){\left|\right|}_{F}^{2}$$
where *W* is a weight matrix, $w_{ij}=1$ if $y_{ij}$ is labeled, namely, interacting or non-interacting; otherwise, $w_{ij}=0$.

Step 3: Applying Tikhonov regularization to avoid overfitting of *A* and *B* to training data: (26)$$\arg\min_{A,B}||W\cdot \left(Y-AB^{T}\right){\left|\right|}_{F}^{2}+\lambda_{\ell }{\left(||A|\right|}_{F}^{2}+{\left||B|\right|}_{F}^{2})$$
where $\lambda_{\ell }$ is a regularization coefficient.

Step 4: Representing drugs similarity $S_{d}$ as approximation of corresponding two drug feature vectors:(27)$$S_{d}\approx AA^{T}$$

Similarly, target similarity $S_{t}$ can be represented as:(28)$$S_{d}\approx BB^{T}$$

Step 5: Linearly combing multiple similarity:(29)$$\begin{array}{c} S_{d}=\sum_{i=1}^{n}w_{d}^{i}S_{d}^{i}\\ S_{t}=\sum_{j=1}^{m}w_{t}^{j}S_{t}^{j}\\ s.t.|w_{d}\left|=\right|w_{t}|=1 \end{array}$$
where $w_{d}=\left(w_{d}^{1},w_{d}^{2},.,w_{d}^{n}\right)$ and $w_{t}=\left(w_{t}^{1},w_{t}^{2},.,w_{t}^{m}\right)$. $w_{d}^{i}$ and $w_{t}^{j}$ are weights from multiple similarity matrices of drugs and targets, respectively.

Step 6: Developing the entire objective function and scoring unknown drug-target pairs:(30)$$\begin{array}{c} \arg\min_{A,B}||W\cdot \left(Y-AB^{T}\right){\left|\right|}_{F}^{2}+\lambda_{\ell }{\left(||A|\right|}_{F}^{2}+{\left||B|\right|}_{F}^{2})+\lambda_{d}\left|\right|\sum_{i=1}^{n}w_{d}^{i}S_{d}^{i}-AA^{T}{\left|\right|}_{F}^{2}\\ \;\;\;\;\;\;\;\;\;\;\;\;\;\;\;\;\;\;+\lambda_{t}\left|\right|\sum_{i=1}^{m}w_{t}^{j}S_{t}^{j}-BB^{T}{\left|\right|}_{F}^{2}+\lambda_{w}\left(|\right|w_{d}{\left|\right|}_{F}^{2}+\left|\right|w_{t}{\left|\right|}_{F}^{2})\\ \;\;\;\;\;\;\;\;\;\;\;\;\;\;\;\;\;\;s.t.\;\;\;|w_{d}\left|=\right|w_{t}|=1 \end{array}$$
where $\lambda_{d}$, $\lambda_{t}$, and $\lambda_{w}$ are regularization coefficients.

The model can be solved with alternating least squares algorithm.

Step 7: Computing the interaction probabilities for unknown drug-target pairs:(31)$$Y_{p}=AB^{T}$$

#### 4.3.4. NRLMF

Liu et al. [61] designed a Neighborhood Regularized-based Logistic Matrix Factorization method (NRLMF) to model the probability of a drug interacting with a target. NRLMF first model the interaction probability $p_{ij}$ between a drug $d_{i}$ and a target $t_{j}$ based on logistic matrix factorization:(32)$$p_{ij}=\frac{\exp(a_{i}b_{j}^{T})}{1+\exp(a_{i}b_{j}^{T})}$$
NRLMF then minimized the following objective function to calculate the interaction probabilities for unknown drug-target pairs by placing spherical Gaussian priors on $a_{i}$ and $b_{j}$:(33)$$\begin{array}{c} \min_{A,B}\sum_{i=1}^{m}\sum_{j=1}^{n}\left(1+cy_{ij}-y_{ij}\right)\log\left[1+\exp\left(a_{i}b_{j}^{T}\right)\right]-cy_{ij}a_{i}b_{j}^{T}+\frac{\lambda_{d}}{2}{\left||A|\right|}_{F}^{2}+\frac{\lambda_{t}}{2}{\left||B|\right|}_{F}^{2} \end{array}$$
where $\sigma_{d}^{2}$ and $\sigma_{d}^{2}$ are used to control Gaussian distribution variances, $\lambda {}_{d}=\frac{1}{\sigma_{d}^{2}}$ and $\lambda {}_{t}=\frac{1}{\sigma_{t}^{2}}$, $\left||A\right|{|}_{F}$ and $\left||B\right|{|}_{F}$ denote the Frobenius norm of A and B, respectively.

The final objective function can be described as: (34)$$\begin{array}{c} \min_{A,B}\sum_{i=1}^{m}\sum_{j=1}^{n}\left(1+cy_{ij}-y_{ij}\right)\ln\left[1+\exp\left(a_{i}b_{j}^{T}\right)\right]-cy_{ij}a_{i}b_{j}^{T}+\frac{1}{2}tr\left[A^{T}\left(\lambda_{d}I+\alpha L^{d}\right)A\right]+\frac{1}{2}tr[B^{T}\left(\lambda_{t}I+\alpha L^{t}\right)B \end{array}$$

#### 4.3.5. DNILMF

Hao et al. [62] extended NRLMF and proposed a Dual-Network integrated Logistic Matrix Factorization method (DNILMF). DNILMF first calculated the interaction probabilities for unknown drug-target pairs:(35)$$P=\frac{\exp(\alpha AB^{T}+\beta S_{d}AB^{T}+\gamma AB^{T}S_{t})}{1+\exp(\alpha AB^{T}+\beta S_{d}AB^{T}+\gamma AB^{T}S_{t})}$$
DNILMF then computed the final interaction scores by maximizing the following objective function:(36)$$\begin{array}{c} \max_{A,B}\sum_{i,j}(cY\circ Z-\left(1+\mathrm{cY}-Y\right)\circ \ln[1+\exp(Z)])-\frac{\lambda_{A}}{2}{\left||A|\right|}_{F}^{2}-\frac{\lambda_{B}}{2}{\left||B|\right|}_{F}^{2} \end{array}$$
where $Z=\alpha AB^{T}+\beta S_{d}AB^{T}+\gamma AB^{T}S_{t}$, ∘ denotes the Hadamard product.

### 4.4. Deep Learning

#### 4.4.1. DeepDTIs

Wen et al. [63] used Deep Belief Network (DBN) to infer potential DTIs without classifying each target into different classes. DeepDTIs identified novel DTIs through three steps:

Step 1: Choosing the most simple and common features to describe drugs and targets: representing chemical compounds with extended connectivity fingerprints and targets with protein sequence composition descriptors.

Step 2: Abstracting feature representations based on DBN. DBN used by DeepDTIs consisted of five layers: the first layer (the input layer) is the calculated features, the second, third and fourth layer are the hidden layers, and the last layer is output layer.

Suppose that *x* is training sample, DeepDTIs modeled the joint probability distribution between *x* and *l* hidden layers based on DBN:(37)$$P\left(x,h^{1},h^{2},.,h^{l}\right)=(\prod_{k=0}^{l-2}P\left(h^{k}|h^{k+1}\right))P\left(h^{l-1},h^{l}\right)$$
where $x=h^{0}$, $P(h^{k-1}|h^{k})$ is a visible-hidden conditional probability distribution at level *k*, $P(h^{l-1},h^{l})$ is the visible-hidden joint probability distribution in the top level.

Step 3: Building a classification model with known label DTIs.

#### 4.4.2. EENN

Gao et al. [64] developed an End-to-End Neural Network (EENN) model to identify DTI candidates directly from raw chemical structures and amino acids sequences. EENN contained four parts: describing drugs and proteins based on related biological information, projecting drugs and proteins into dense vector spaces by integrating graph-based convolutional neural network and long short-term memory recurrent neural networks, forming the context matrix for drugs and protein with attentive pooling network and computing weighted sums of the context matrix, and predicting the interaction probabilities for unknown drug-target pairs based on inference with siamese network. The details are shown in Figure 11.

#### 4.4.3. Stacked Autoencoder

Wang et al. [65] designed a novel computational model to find possible DTIs combining stacked autoencoder in deep learning models. The proposed method can automatically screen hidden information from raw data and select highly representative features based on iterations of multiple layers.

The method can be grouped into four parts: describing each DTI (sample) based on 881 chemical structures of drugs and the position-specific scoring matrix related to protein, reconstructing features with stack autoencoder, classifying unknown drug-target pairs with random forest classifier, and predicting labels for test samples. The details are shown in Figure 12.

In step 2, Wang et al. first encoded the training sample $X\in R^{d_{0}}$ into the hidden representation $H\in R^{d_{1}}$ by the mapping $f_{c}$:(38)$$H=f_{c}\left(X\right)=J_{c}\left(W_{1}^{T}X+b_{1}\right)$$
where $J_{c}$ is the activation function, $W_{1}$ and $b_{1}$ are weighted parameters $W_{1}\in R^{d_{0}\times d_{1}}$ and bias vector $b_{1}\in r^{d_{1}}$, respectively. The representation of the hidden layer *H* is then mapped into the output layer $Z\in R^{d_{0}}$ by the mapping $f_{d}$:(39)$$Z=f_{d}\left(H\right)=J_{d}\left(W_{2}^{T}H+b_{2}\right)$$
where $J_{d}$ is the activation function, $W_{2}$ and $b_{2}$ is weighted parameters $W_{2}\in R^{d_{0}\times d_{1}}$ and bias vector $b_{2}\in r^{d_{0}}$, respectively. The parameters can be learned by minimizing the following loss function:(40)$$\begin{array}{c} \Theta \left(X,Z\right)=\Theta_{r}\left(X,Z\right)+0.5\tau (||W_{1}{\left|\right|}_{2}^{2}+\left|\right|W_{2}{\left|\right|}_{2}^{2}) \end{array}$$
where $\Theta_{r}\left(X,Z\right)$ and τ are the reconstruction error and the weight decay cost, respectively. The hidden layer learned the features and reduced the dimension of original data by mapping. The highest hidden layer of autoencoder can be used as the features of raw data extracted by the stacked autoencoder.

### 4.5. Other Methods

#### 4.5.1. RBM

Wang et al. [66] learned associated probabilities of unknown drug-target pairs using a two-layer restricted Boltzmann machine (RBM) where visible units encoded types of DTIs and hidden units represented latent features of DTIs. Figure 13 describes the details.

#### 4.5.2. NetCBP

Chen et al. [67] exploited a semi-supervised learning-based prediction model (NetCBP) combined with network consistency. NetCBP assumed that there existed coherent interactions between drugs ranked based on their correlations to a query drug and targets ranked based on their correlations to the hidden targets of the query drug, and then designed a learning model to maximize the rank coherences relevant to known DTIs. The details are described in Figure 14.

## 5. Discussion

Drug repurposing involves various computational methods [1,3]. Of these techniques, DTI inference is one of the most important foundations [68,69]. In this paper, we summarized data sources and related representation involved in DTI prediction. We mainly introduced two classes of typical computational models, network-based methods and machine learning-based methods. These two types of models are applied to target proteins without any known 3D structure information and obtained effective prediction performance [52,70]. More importantly, almost all the methods can further infer novel DTIs for drugs interacting with at least one target protein [4]. Furthermore, some algorithms can effectively identify DTI candidates for new drug molecules which have no associated information with targets by combining with drug similarity, target similarity, and DTIs [4,52,71]. However, there are a few limitations to solve.

Network-based methods are limited to application because DTI data are severely imbalanced in the relevant dataset and there are many more unknown drug-target pairs than DTIs in DTI network [4,72,73]. For example, the interactions in ion channel dataset provided by Yamanishi et al. [9] should be 210×204= 42,840, however, the actual interaction is 1467. More importantly, a DTI network usually contains several isolated subnetworks, where network-based models are unable to find new association information for orphan drugs (or targets) which have not any known interaction data in the DTI network [4,70]. Finally, most of the network-based methods are biased toward the drugs (or targets) which tend to interact with more targets (or drugs) [4,73]. Therefore, network-based methods should be further exploited to solve these problems in the future.

Machine learning-based methods obtained good improvement in the process of DTI prediction. Table 2 and Table 3 illustrate the performances of some machine learning-based methods from Refs. [52,61]. Table 2 lists AUC and AUPR values provided by Mei et al. [52] for KRM, BLM, RLS${}_{GIP}$, and BLM-NII. These methods are BLM-based methods. The results show that BLM-NII obtained better performance than other BLM-based methods and prove that neighbor-based interaction-profile helps to predict new DTIs.

Table 3 lists the AUC and AUPR values provided by Liu et al. [61] for NetLapRLS, BLM-NII, WNN-GIP, KBMF2K, and NRLMF where NetLapRLS and WNN-GIP are regularized least squares-based methods, BLM-NII is BLM-based method, and the remaining are matrix factorization-based methods. The minor difference of BLM-NII in Table 2 and Table 3 may be caused by different experimental settings.

Matrix factorization models obtain better performance for DTI identification [59,60,61,62,74]. However, this type of method has more parameters to set and is sensitive to parameters [73]. Although RLS-WNN cannot outperform matrix factorization methods, it is relatively much faster and more robust to parameter selection [73,75]. BLMs can efficiently process many fewer unknown DTIs, and thus they exhibit much lower complexity than global algorithms. Furthermore, BLMs are usually fast and memory-efficient techniques when the dataset used is larger [52,73]. Nevertheless, BLMs cannot deal with the situation that both drugs and targets are not included in the training dataset unless integrated with other methods, for example, BLM-NII [74]. Deep learning-based methods obtained better improvement because of their powerful representation learning ability and are one powerful models for DTI prediction [63,65,76,77].

In summary, although various machine learning-based methods have been already proven to be effective for DTI identification, various challenges still remain.

(i) Most of the supervised learning methods are limited to the negative sample selection problem because there are not experimental validated non-DTI data. Therefore, this type of method can only randomly select negative DTI data from unknown associated drug-target pairs, however, these selected negative samples may contain positive DTIs, which severely affects classification performance and generalization ability of models [4,10,56,71,73,74].

(ii) Machine learning-based prediction models are usually built and evaluated with an excessively simplified experimental setting. Such settings may wander from the real case and produce over fitting results [4,74]. Especially, most of the machine learning-based models simply regard DTI as an on-off association and do not consider other key factors like quantitative affinities and molecule concentrations [56,74]. Pahikkala et al. [56] have illustrated that at least four factors may result in highly positive predictive results when building and measuring supervised machine learning-based methods: experimental setting, evaluation data set, problem formulation and evaluation setup. Therefore, DTI identification should be modeled as a rank or regression problem rather than a binary classification problem [74].

(iii) When predicting possible DTIs based on binary classification, the classification accuracy is biased because the results are from the simple average of two different classification models, which are constructed based on drugs and targets, respectively [4].

(iv) Most of the machine learning-based methods have “poor interpretability” properties, therefore, it is difficult to understand potential drug mechanism of action from a pharmacology viewpoint [74].

Although semi-supervised learning methods overcame the negative sample selection limitation by making use of the unlabeled data, it still cannot solve the problem of classifier combination [4].

## 6. Conclusions and Further Research

In this section, we attempt to provide some suggestions of further research on how to improve DTI prediction performance.

### 6.1. Heterogeneous Data Integration

Most models incorporate chemical and genomic information, in addition, previous works have utilized pharmacological or phenotypic information, such as side-effects data, gene expression information, and some associated data. These data represent different natures of drugs and targets and can boost prediction accuracy if used concurrently. However, most existing models are limited to homogeneous information and cannot be directly applied to heterogeneous networks.

Heterogeneous data sources give diverse information and help find possible DTIs from a multi-view perspective. To the best of our knowledge, for instance, some genes coding proteins (targets) are tightly associated with some diseases and the therapeutic effects of the drugs on these diseases reflect their biological activities to these targets. Therefore, integrating with various heterogeneous data sources, such as gene-disease association network, drug-disease association network, metabolic network associated to specific diseases, can potentially improve the accuracy and thus provide new insights.

Although several network-based strategies incorporate heterogeneous data source and derive the associated scores through network diffusion method, most existing models have some limitations and fail to give satisfactory integration paradigms: first, the noise and high-dimensionality natures of biological data easily cause predicted bias. Moreover, some network-specific information may be lost in the process of integrating multiple different networks into a single network, since edges from multiple heterogeneous networks are mixed indiscriminately in such process. Therefore, designing appropriate models to incorporate multiple relevant heterogeneous data sources still remains an open problem.

### 6.2. Reliable Negative Sample Selection

There exist parts of known DTIs (positive samples) and massive unknown drug-target pairs in existing DTI datasets. In addition, there are not experimental validated non-DTIs (negative samples) so that most of the supervised classification algorithms have no choice but to randomly select unlabeled drug-target pairs as negative samples. However, this part of randomly selected negative samples, in fact, may well contain positive DTIs, thereby severely confusing the classification accuracy of supervised-learning techniques. Therefore, although extracting positive drug-target pairs from unconfirmed data is an urgent task, designing an effective method to screen negative DTIs is more challenging [10]. To the best of our knowledge, positive-unlabeled learning [71,78,79] can learn high-quality positive samples and reliable negative samples from the unlabeled data and may be one effective way to select strong negative DTIs.

### 6.3. Noncoding RNAs as Targets

It is worth mentioning to consider noncoding RNAs as drug targets. Noncoding RNAs [80,81] (nc RNAs) are another new class of targets. ncRNAs can control gene expression and affect disease progression, which makes them targets in the process of drug research and discovery. ncRNAs consist of multiple functionally important RNAs including transfer RNA (tRNA), microRNA, intronic RNA, ribosomal RNAs (rRNA), long noncoding RNA, and repetitive RNA. Each class of RNA has different endogenous functions, which provides many opportunities for drug discovery and design.

ncRNAs have been considered as targets and obtained increasing attention. For example, microRNAs have been well-reviewed to be therapeutically targeted candidates [82,83]. Both microRNA mimics and inhibitors are being designed against targets and tested in clinical trials. For instance, the drugs BMN 044/ PRO044, BMN 045/ PRO045, BMN 053/ PRO053, SRP-4053, and SRP-4053 can be used to therapy duchenne muscular dystrophy (DMD) by targeting dystrophin pre-mRNA [81]. Recently, the research on targeting of repetitive RNAs, intronic RNAs, and miRNAs are advanced, however, long ncRNAs, which are regarded as a challenging class of possible drug targets, will be further focused upon.

The researchers exploited several ncRNA databases, such as NONCODE (http://www.noncode.org/), Noncoding RNA database (http://biobases.ibch.poznan.pl/ncRNA/), RNAcentral (http://www.rnacentral.org/), miRBase (http://www.mirbase.org/), lncrna (http://www.lncrnadb.org/), and Ensembl (http://asia.ensembl.org/info/genome/genebuild/ncrna.html). These databases provide a mass of ncRNA information and help us predict underlying associations between drugs and ncRNAs. Especially, NONCODE gives various ncRNA data excluding tRNAs and rRNAs for 17 species. The noncoding RNA database contains more than 30,000 individual sequence and function information of ncRNAs from 99 species of Archaea, Bacteria, and Eukaryota.

### 6.4. Environmental Factors and Genetic Factors

Various studies have reported that associations between genetic factors (GFs) and environmental factors (EFs) can greatly influence phenotypes and diseases [84,85]. The computational modeling of GF-EF interaction prediction considerably enriches our knowledge on the mechanisms of GF-EF interactions. For instance, drugs, one class of important EFs, have been revealed to interact with targets (GFs) [84,85]. Qiu et al. [85] suggested that miRNA biomarker signatures of drugs could be applied to evaluate the effects of cancer treatments. Therefore, the analysis and identification of interactions between drugs and genetic factors could help infer novel indications for FDA approved drugs.

### 6.5. Deep Learning

In the era of big data, large quantities of biological data are dramatically increasing. The availability of these datasets have promoted the development of various modeling approaches [63,76]. Deep learning approach is one type of representation-learning method that can be applied to deal with complex works with heterogeneous and high-dimensional datasets. The accumulation of massive drug and target data provides quantities of biomedical features and accelerates the application of deep learning on DTI prediction [77,86]. Although several deep learning methods [63,64,65] are used to identity possible DTIs, there remains many challenges in interpreting deep learning results, such as selecting appropriate deep architectures and model parameters, solving with small samples and high-dimensional nature of the datasets. Therefore, building an appropriate deep model may be one of efficient ways to improve DTI prediction performance.

### 6.6. Sparse Representation

DTI data in DTI network are sparse and imbalanced. There is a small quantity of DTIs and abundant unknown drug-target pairs. For example, in the datasets provided by Yamanishi et al. [9], the number of DTIs are 2926, 1476, 635, and 90 between 445, 210, 223, and 54 drugs and 664, 204, 95, and 26 target proteins, respectively, from enzymes, ion channels, GPCRs, and nuclear receptors. The ratio of known DTIs to all drug-target pairs is 0.0099, 0.0345, 0.03, and 0.0641, respectively. The dataset provided by Wen et al. [63] contains only 6262 DTIs among possible 2,146,240 (1412×1520) drug-target pairs from 1412 drugs and 1520 targets, and the ratio of known DTIs to all drug-target pairs is 0.0029. More importantly, DTI prediction must be solved in small samples with high dimension natures of drugs and target information. Sparse representation can automatically discriminate various classes and provides a simple and effective ways of rejecting any invalid test samples not from any class in the training set, and thus reduces data dimension and computational cost [87]. Therefore, sparse representation-based methods may be further applied to DTI prediction.

### 6.7. Types of DTI

Different types of DTIs help us understand the molecular mechanism of drug action. Although the existing methods have achieved promising performance, the majority of them can only infer the binary interaction between a drug and a target, but cannot detect distinct types of interactions. However, the interactions between drugs and targets generally have different meanings, for example, direct interactions produced by protein-ligand binding and indirect interactions caused by either changed expression levels of a target protein or active metabolites induced by a drug [16,66]. In addition, DTIs can be annotated by different drug modes of action, such as activation and inhibition [17]. Therefore, how to use various biological data to identify different types of DTIs may be a challenging problem.

### 6.8. Personalized Medicine

The ultimate goal of DTI identification is to provide treatment clues for patients, especially for cancer patients. However, it is inappropriate to simply use one or a few drugs for all the patients [88]. Therefore, computational methods should be used to mine personalized drugs by integrating cancer-related network, drug-drug interaction network, protein-protein interaction network, metabolic network, and so on. Fusing this important information and novel network-based models, researchers may find some valuable drug discovery strategies. In addition, computational models could be applied to predict personalized drug targets, drug effects and resistances for cancer treatment, and infer personalized cancer risk for healthy individuals [89,90]. Therefore, performing personalized medicine based on DTI identification may be a topic of further research.

## Figures and Tables

**Figure 1 molecules-24-01714-f001:**
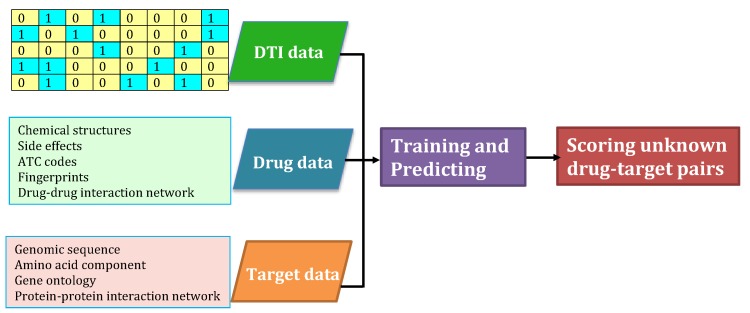
The flowchart of standard drug-target interactions (DTI) identification models.

**Figure 2 molecules-24-01714-f002:**
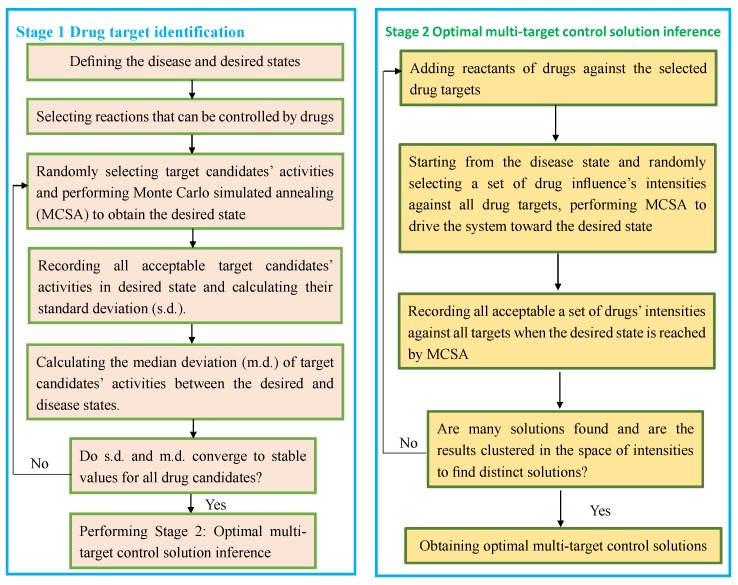
The flowchart of multiple target optimal intervention solutions (MTOI).

**Figure 3 molecules-24-01714-f003:**
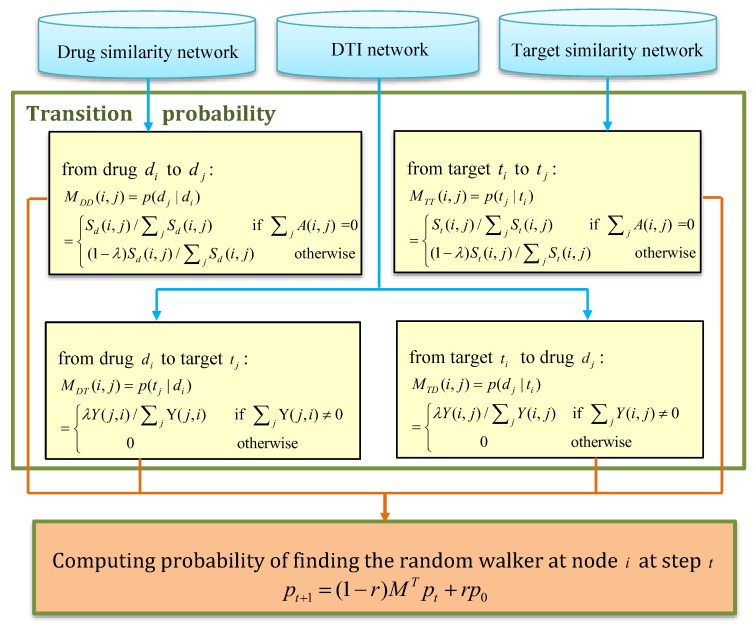
The flowchart of Network-based Random Walk with Restart on the Heterogeneous network (NRWRH).

**Figure 4 molecules-24-01714-f004:**
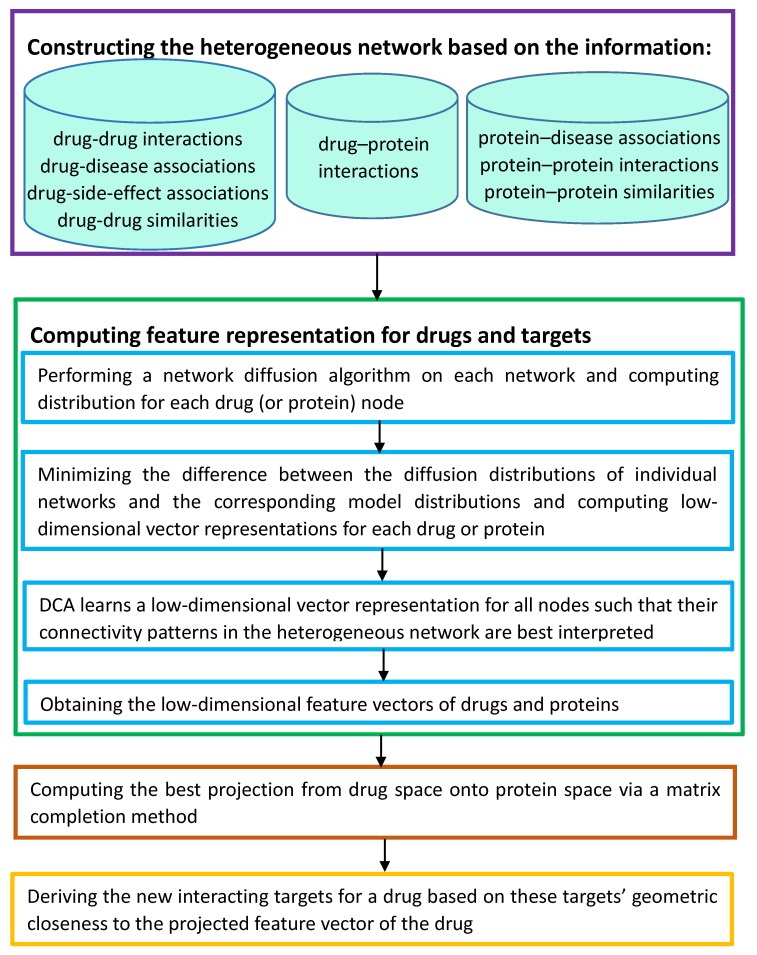
The flowchart of a novel Network integration pipeline for DTI prediction (DTINet).

**Figure 5 molecules-24-01714-f005:**
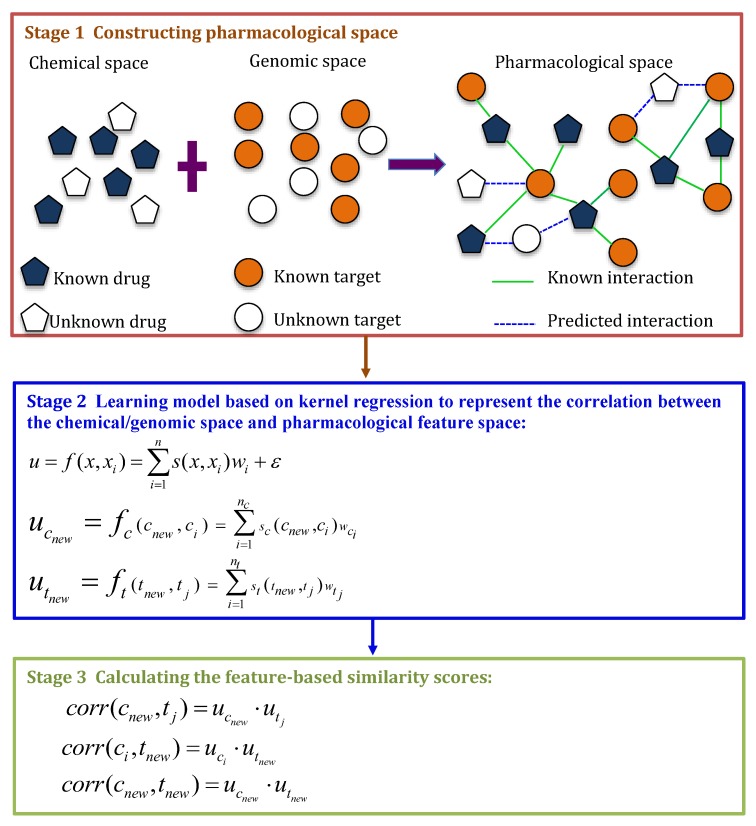
The flowchart of Kernel Regression Method (KRM).

**Figure 6 molecules-24-01714-f006:**
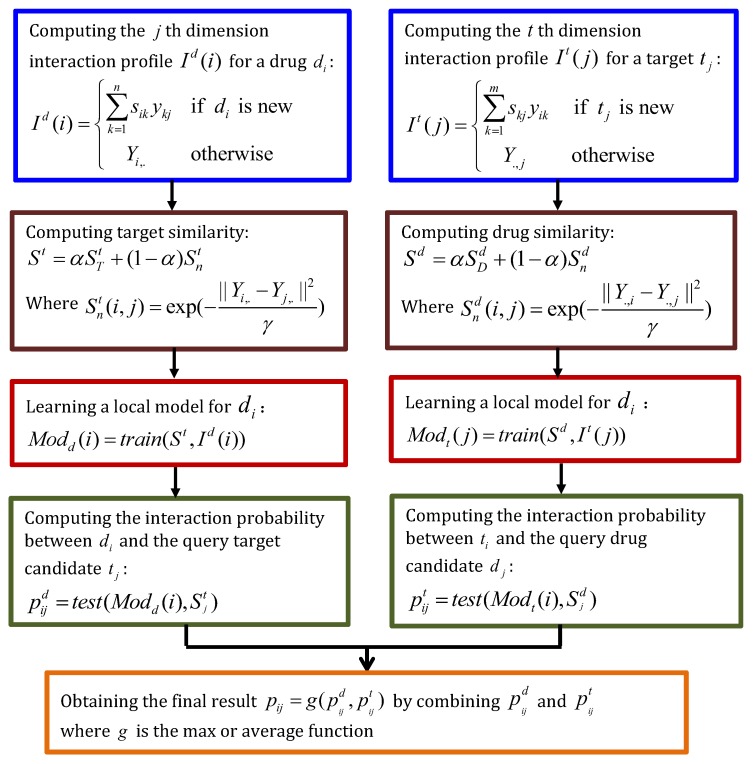
The flowchart of BLM with neighbor-based interaction-profile inferring (BLM-NII).

**Figure 7 molecules-24-01714-f007:**
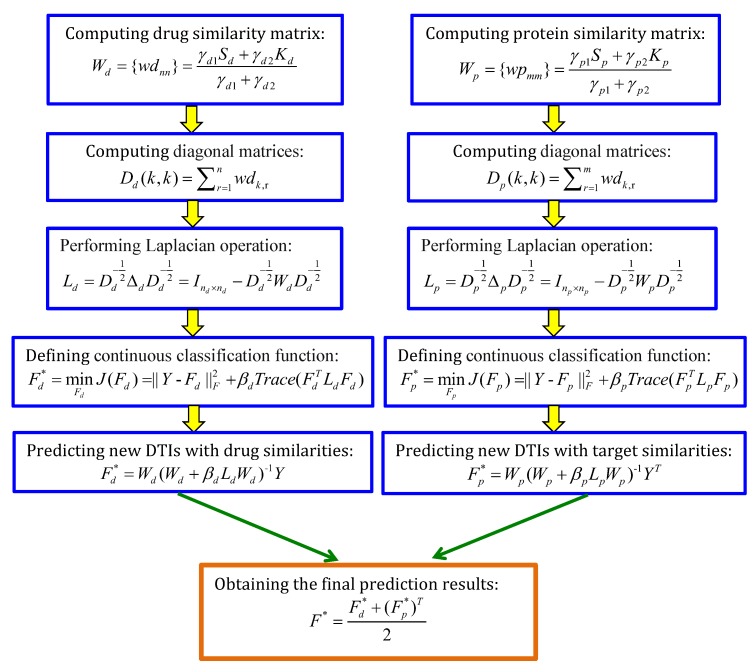
The flowchart of Laplacian regularized least square (LapRLS) incorporating DTI network (NetLapRLS).

**Figure 8 molecules-24-01714-f008:**
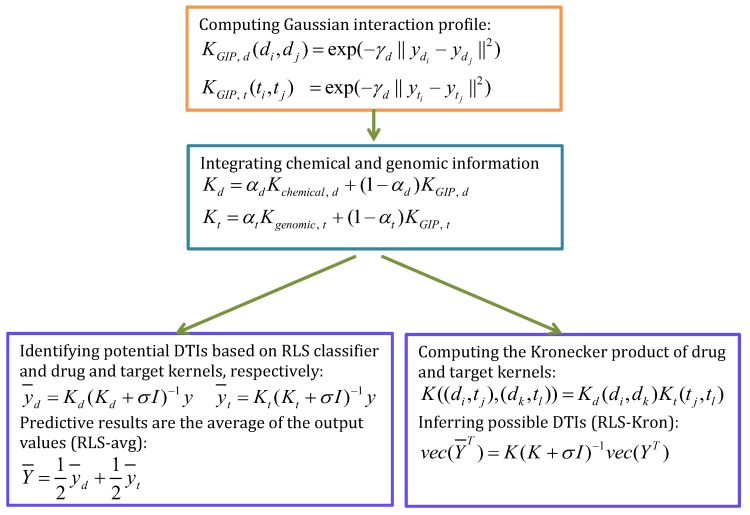
The flowchart of regularized least squares (RLS)${}_{GIP}$.

**Figure 9 molecules-24-01714-f009:**
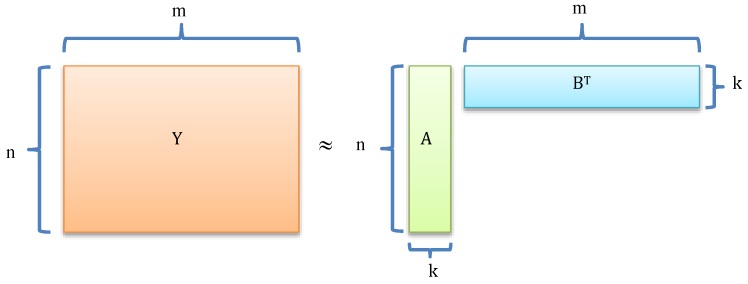
The flowchart of DTI identification methods based on matrix factorization.

**Figure 10 molecules-24-01714-f010:**
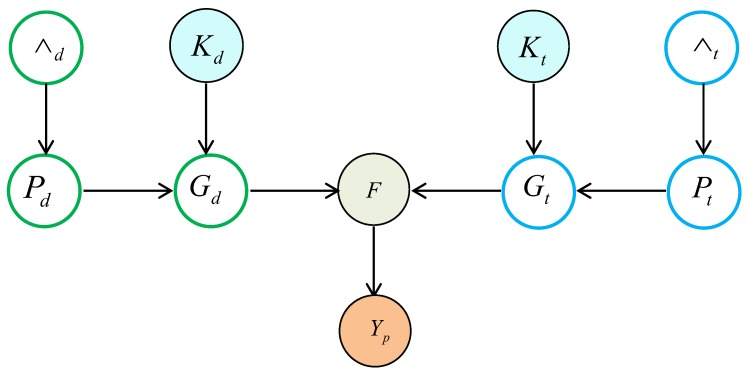
The flowchart of Kernelized Bayesian Matrix Factorization with twin Kernels (KBMF2K).

**Figure 11 molecules-24-01714-f011:**
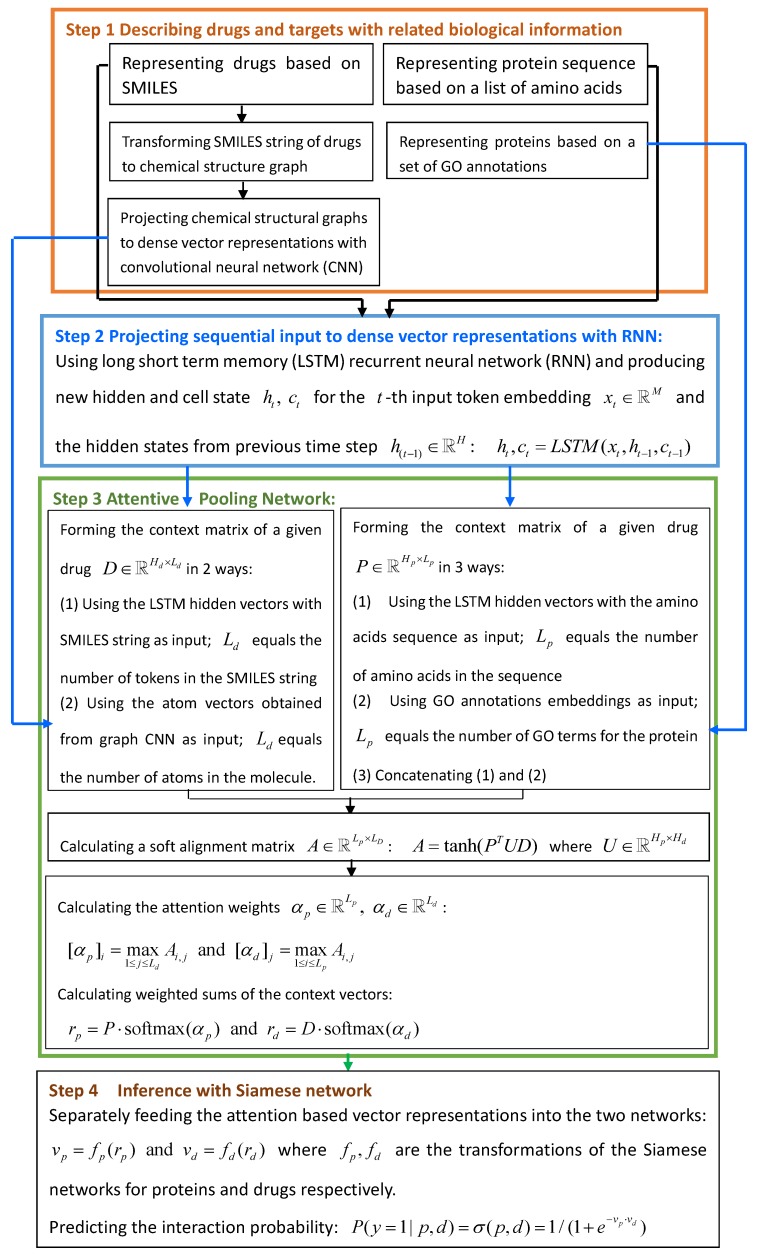
The flowchart of End-to-End Neural Network (EENN).

**Figure 12 molecules-24-01714-f012:**
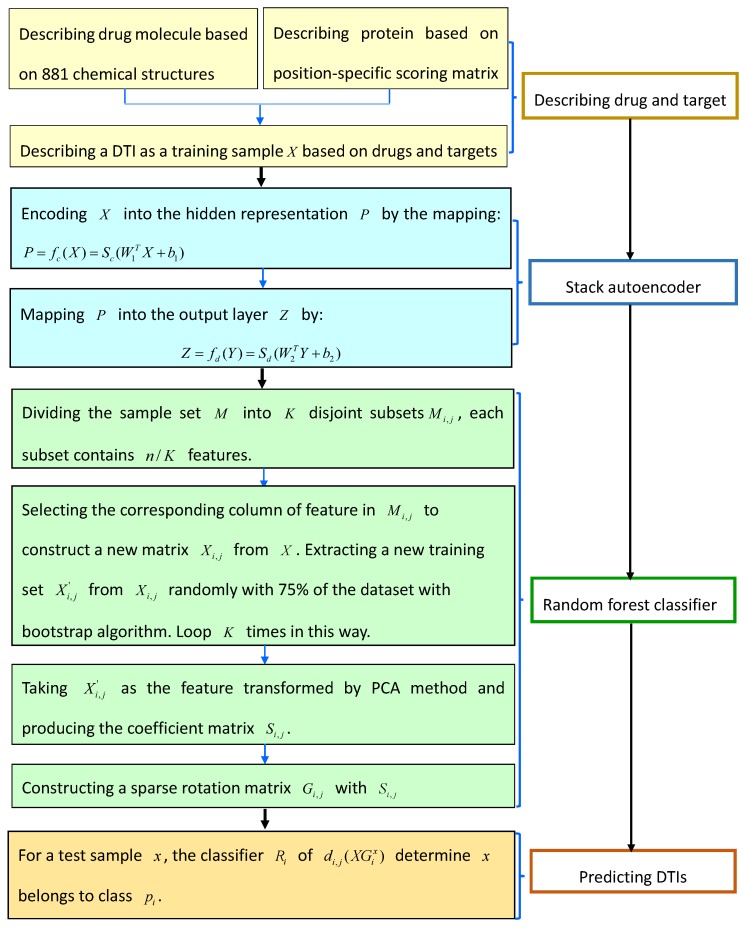
The flowchart of stacked autoencoder.

**Figure 13 molecules-24-01714-f013:**
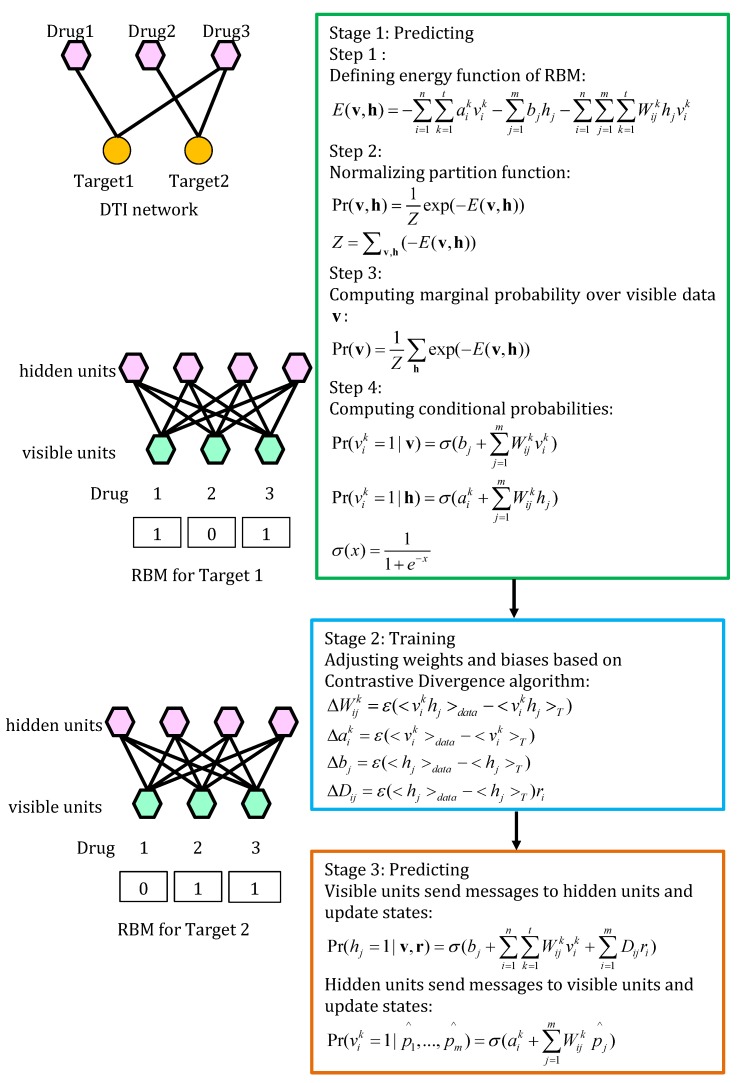
The flowchart of restricted Boltzmann machine (RBM).

**Figure 14 molecules-24-01714-f014:**
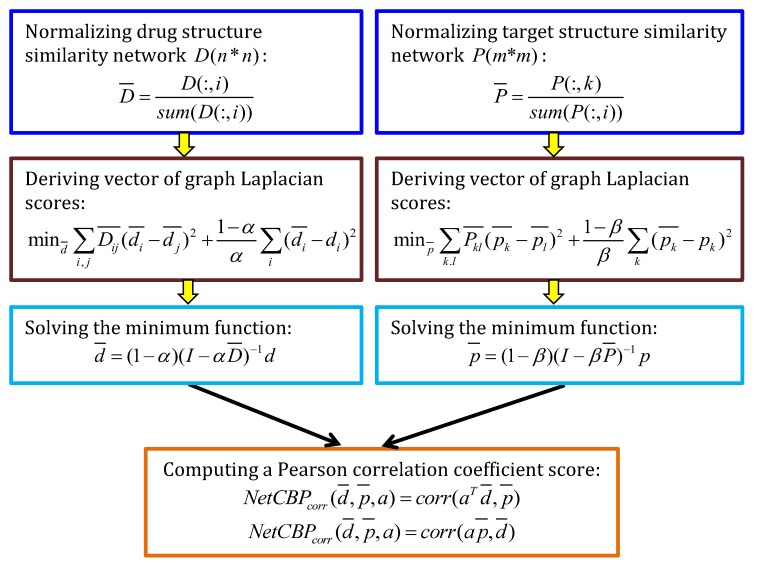
The flowchart of NetCBP.

**Table 1 molecules-24-01714-t001:** Datasets provided by Yamanishi et al. [9].

Dataset	Drugs ($n_{d}$)	Targets ($n_{t}$)	Interactions
enzyme	445	664	2926
ion channel	210	204	1476
GPCRs	223	95	635
nuclear receptor	54	26	90

**Table 2 molecules-24-01714-t002:** Performance comparison of BLM-based methods [52].

		**AUC**		
**Dataset**	**KRM**	**BLM**	**RLS** ${}_{\mathrm{GIP}}$	**BLM-NII**
Enzyme	86.4	97.6	97.8	98.8
Ion Channel	81.9	97.3	98.4	99.0
GPCR	76.5	95.5	95.4	98.4
Nuclear Receptor	74.9	88.1	92.2	98.1
		**AUPR**		
**Dataset**	**KRM**	**BLM**	**RLS** ${}_{\mathrm{GIP}}$	**BLM-NII**
Enzyme	6.30	83.3	91.5	92.9
Ion Channel	17.2	78.1	94.3	95.0
GPCR	10.9	66.7	79.0	86.5
Nuclear Receptor	17.1	61.2	68.4	86.6

**Table 3 molecules-24-01714-t003:** Performance comparison of different types of prediction models [61].

		**AUC**			
**Dataset**	**NetLapRLS**	**BLM-NII**	**WNN-GIP**	**KBMF2K**	**NRLMF**
Enzyme	97.2	97.8	96.4	90.5	98.7
Ion Channel	96.9	98.1	95.9	96.1	98.9
GPCR	91.5	95.0	94.4	92.6	96.9
Nuclear Receptor	85.0	90.5	90.1	87.7	95.0
		**AUPR**			
**Dataset**	**NetLapRLS**	**BLM-NII**	**WNN-GIP**	**KBMF2K**	**NRLMF**
Enzyme	78.9	75.2	70.6	65.4	89.2
Ion Channel	83.7	82.1	71.7	77.1	90.6
GPCR	61.6	52.4	52.0	57.8	74.9
Nuclear Receptor	46.5	65.9	58.9	53.4	72.8

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
