# Peer review of "Revealing Drug-Target Interactions with Computational Models and Algorithms"

_molecules, 2019, doi:10.3390/molecules24091714_

Round 1
Reviewer 1 Report
The review concerns the discussion and description of available computational models for DTI identification focusing mainly on network-based algorithms and machine learning-based methods. The authors also discussed potential avenues for DTI prediction accuracy.
The paper is overall interesting and accurate in the description of the main approaches and algorithms for Network and machine learning methods. However, the discussion section is too short and unbalanced with respect the MLM methodological description.
I suggest expanding this section also citing some applicative papers that used the DTI approaches described in the manuscript.
Other suggested revisions:
English must be checked and corrected by native speakers; infact, the both introduction and the discussion section presents not clear sentences that prevent following the whole meaning.
Minor revisions:
· Page 1 line 24-25:
· “DTI identification, aiming to find potential targets (drugs) for the existing drugs (targets), has been
· 25 an important step in drug repositioning.”
· I suggest removing the brackets and using the / instead. Otherwise, it could be misunderstood.
· Section 1- repositories and databases: include also ZINC database in the list.
· Section 4: DISCUSSION, page 24; “drug molecules without known 3D structures” it not clear…specify better..do you mean that thare no chemical compounds known acting as drugs for a specific target? Or what exactly?
· All the authors cited in the text of the review must be reported with the first character as capitol letter (there are many…check through all the paper)
· Yamanish instead of yamanishi…etc.
· Reference 36, is bad reported..the author surnames are missing
Author Response
Dear Reviewer,
We really appreciate you for your valuable comments. Following your comments, we have thoroughly updated the manuscript. We provide our responses in a “point-by-point” manner and list them in the attachment (for better readability, our responses are in blue color while the original comments are in black).
We are very thankful for your suggestions!
With best regards!
Yours sincerely,
Lihong Peng
Hunan University of Technology
Zhuzhou, Hunan, China

Reviewer 2 Report
Recommendation: Publish in Molecules after minor revisions
Comments:
Exploring potential drug-target interactions (DTIs) is among hot areas in drug discovery, in particular for drug repurposing and off-target predictions. In this review, the authors nicely summarized the existing computational methods, database and algorithms for prediction of potential DTIs. In addition, the author provided an insightful discussion about the strengths and limitations of different methods, as well as some suggestions for the future directions in this field. As such, I believe this paper would benefit to a broad audience in drug discovery.
However, some concerns are suggested before publication in Molecules.
1. How DTIs are identified in different DTI database? Is it feasible to integrate different DTI associations from different DTI database?
The authors talked about a variety of available DTI database in Section 1, and also stressed the significance of integration of heterogeneous data. However, the readers may consider how the DTIs are identified in different DTI database. As we know, there might exist multiple measurements, i.e. the binding affinity, IC50, EC50, etc, or multiple values for the same measurement for a single drug-target interaction. How does different database (such as DrugBank, SuperTarget, …) convert multiple experimental data to an identified DTI? Can the data from different database be feasibly merged?
It would be nice if the authors include these information in Section 1.
2. The lack of benchmark study between different DTI prediction algorithms.
The authors nicely discussed multiple DTI prediction algorithms. But other than the algorithms themselves, people might be more interested in the comparison of the performance for those algorithms. It would be great if the author can summarize a case study or a summarized table to show the performance of different DTI prediction algorithms.
3. Are there any on-line tools/ web-service for DTI prediction? This is might be useful from the practitioners’ point of view.
4. In the Keywords, “machine-based methods” should be corrected as “machine learning-based methods”
5. The font in Figure 2 is too small. The author may consider change one-column figure to a two-column one.
6. On Page 24 Line 399-400, the author argued “…. the matrix factorization models may be the best DTI identification methods”. But why? The author should add more rationales or references for this conclusion, in particular why the matrix factorization models are better than other deep learning methods.
Author Response

(The authors gave the same response as above.)

Round 2
Reviewer 1 Report
The revisions made are good and the paper is now suitable to be published in this revised form